# Antibiotic Treatment of Infections Caused by AmpC-Producing Enterobacterales

**DOI:** 10.3390/pharmacy12050142

**Published:** 2024-09-21

**Authors:** Gianpiero Tebano, Irene Zaghi, Monica Cricca, Francesco Cristini

**Affiliations:** 1Infectious Diseases Unit, Ravenna Hospital, AUSL Romagna, 48100 Ravenna, Italy; 2Department of Infectious Diseases, University Hospital of Galway, H91 Galway, Ireland; irene.zaghi@gmail.com; 3Unit of Microbiology, The Greater Romagna Area Hub Laboratory, 47522 Cesena, Italy; monica.cricca3@unibo.it; 4Department of Medical and Surgical Sciences (DIMEC), Alma Mater Studiorum, University of Bologna, 40126 Bologna, Italy; francesco.cristini@ausltromagna.it; 5Infectious Diseases Unit, Forlì and Cesena Hospitals, AUSL Romagna, 47121 Forlì and Cesena, Italy

**Keywords:** AmpC enzyme, ampC gene, *Enterobacter cloacae complex*, *Klebsiella aerogenes*, *Citrobacter freundii*, beta-lactams, beta-lactamases, antibiotic resistance

## Abstract

AmpC enzymes are a class of beta-lactamases produced by Gram-negative bacteria, including several Enterobacterales. When produced in sufficient amounts, AmpCs can hydrolyze third-generation cephalosporins (3GCs) and piperacillin/tazobactam, causing resistance. In Enterobacterales, the AmpC gene can be chromosomal- or plasmid-encoded. Some species, particularly *Enterobacter cloacae* complex, *Klebsiella aerogenes*, and *Citrobacter freundii*, harbor an inducible chromosomal AmpC gene. The expression of this gene can be derepressed during treatment with a beta-lactam, leading to AmpC overproduction and the consequent emergence of resistance to 3GCs and piperacillin/tazobactam during treatment. Because of this phenomenon, the use of carbapenems or cefepime is considered a safer option when treating these pathogens. However, many areas of uncertainty persist, including the risk of derepression related to each beta-lactam; the role of piperacillin/tazobactam compared to cefepime; the best option for severe or difficult-to-treat cases, such as high-inoculum infections (e.g., ventilator-associated pneumonia and undrainable abscesses); the role of de-escalation once clinical stability is obtained; and the best treatment for species with a lower risk of derepression during treatment (e.g., *Serratia marcescens* and *Morganella morganii*). The aim of this review is to collate the most relevant information about the microbiological properties of and therapeutic approach to AmpC-producing Enterobacterales in order to inform daily clinical practice.

## 1. Introduction

Antibiotic resistance is a widespread phenomenon and represents a global threat [1]. Antibiotic resistance in Gram-negative pathogens can be caused by a variety of mechanisms, including enzymatic inactivation of the antibiotic, porin mutations, overexpression of efflux pumps, and target modification. Enzymatic inactivation is particularly relevant to beta-lactam antibiotics, which are pivotal agents in treating severe infections [2,3,4,5,6].

Overall, beta-lactam-inactivating enzymes are identified as beta-lactamases [2,6,7,8]. All beta-lactamases can hydrolyze the beta-lactam ring, and this structural modification compromises the activity of the antibiotic. However, beta-lactamases are a vast group of enzymes and can differ significantly with regard to molecular structure, genetic features, regulation of genetic expression, potential for clonal spreading and horizontal transmission, efficacy in hydrolyzing their substrate, range of action, and resistance to different beta-lactamase inhibitors [8,9,10,11].

AmpC enzymes represent a vast and concerning family of beta-lactamases. AmpC enzymes (hereinafter defined for brevity as AmpCs) are found in many critically relevant human pathogens, including many Enterobacterales, *Pseudomonas aeruginosa*, and *Acinetobacter baumannii* [9]. They can compromise the effectiveness of most beta-lactams, leaving few therapeutic options. The treatment of AmpC-producing pathogens can be complex, with many areas of uncertainty and possible pitfalls. Some authoritative reviews about this topic exist in the literature [3], but recent advances have occurred, justifying an updated revision of available data. The aim of this review is to provide a practical overview of the most relevant microbiological properties of AmpC-producing Enterobacterales and the antibiotic treatment of infections caused by these microorganisms, with a special focus on inducible chromosomal AmpCs.

## 2. Microbiological Properties

### 2.1. What Are AmpC Enzymes?

AmpCs are beta-lactamases produced by a wide range of Gram-negative bacteria [9,12]. Like several other beta-lactamases, they are structurally similar to the beta-lactam target, i.e., the penicillin-binding proteins (PBPs). They contain a serine in the active site and belong to class C according to the Ambler structural classification [7] and to group 1 according to the Bush functional classification [2].

The first evidence of AmpC production is dated back to 1940. It appears that the very first bacterial enzyme reported to be able to destroy penicillin was structurally an AmpC, produced by *Escherichia coli*, although it was so named only several years later [13]. Afterwards, several enzymes have been discovered and assigned to the AmpC class according to their molecular structure and target substrate [12].

The AmpC gene carried by Enterobacterales can be chromosomally or plasmid-encoded [3,12]. The first AmpC originally described in 1940 was chromosomally encoded [14,15]. In the late 1980s, the first plasmid-encoded AmpC gene was reported [16], and subsequently, several plasmid-mediated class C beta-lactamases have been discovered worldwide, both in clinical samples [17] and environmental settings [18].

All AmpCs produced by Enterobacterales are able to hydrolyze penicillins, monobactams, and cephalosporins, including ceftaroline and ceftobiprole. The only notable exception is cefepime. AmpC-producing strains remain susceptible to carbapenems and to most newer beta-lactam/beta-lactamase inhibitor combinations (BL-BLICs) (Table 1) [9,12,19,20].

Despite a wide overlap in the spectrum of activity, AmpCs can usually be differentiated phenotypically from extended-spectrum beta-lactamases (ESBLs). ESBLs are inhibited by older beta-lactamase inhibitors (clavulanic acid, tazobactam), do not hydrolyze cephamycins (cefoxitin), and frequently confer resistance to cefepime. AmpCs produced by Enterobacterales are not inhibited by clavulanic acid and tazobactam and confer resistance to cefoxitin; in contrast, they do not efficiently hydrolyze cefepime (Table 1) [4,9,12,21].

### 2.2. How Can AmpC Enzymes Be Classified?

One peculiar characteristic that differentiates AmpCs from other beta-lactamases is that their expression can be either constitutive or inducible [3,9,12]. This can have relevant implications in a clinical setting.

According to this characteristic, AmpCs can be classified into three groups (Figure 1):(1)Inducible chromosomal AmpCs.(2)Non-inducible chromosomal AmpCs.(3)Plasmid-encoded AmpCs.

(1) Inducible chromosomal AmpCs follow a complex mechanism. AmpC production is usually maintained at a low level by the presence of a negative regulatory element named AmpR, which down-regulates the expression of the AmpC gene. AmpR’s activity can be disabled when it is bound by some peptides, which can induce conformational changes; such peptides derive from bacterial cell wall degradation, and their production is increased by beta-lactam antibacterial action. Thus, beta-lactam exposure can determine the derepression of the AmpC gene, with consequent hyperproduction of AmpC, a phenomenon usually called AmpC derepression [22,23,24].

Other molecules are implicated in the regulation of AmpC production, such as AmpD, a cell wall enzyme that removes peptides derived from cell wall disruption, preventing AmpC derepression [25].

AmpC derepression due to beta-lactam exposure is usually a reversible phenomenon; thus, AmpC production can be down-regulated again when the exposure to beta-lactam is discontinued. However, mutations can further occur in regulatory genes, such as ampR or ampD (coding for AmpR and AmpD, respectively), determining a stable (non-reversible) derepression of AmpC production [3,26].

Inducible chromosomal resistance is mainly found in *Enterobacter cloacae complex*, *Klebsiella aerogenes*, and *Citrobacter freundii*, but can be present in many other Enterobacterales, as we will discuss later in more detail [17,27,28,29,30]. When inducible chromosomal AmpC is not derepressed, these bacteria appear phenotypically resistant to amoxicillin/ampicillin, amoxicillin/clavulanic acid, and first-generation cephalosporins but susceptible to ticarcillin, piperacillin, and third-generation cephalosporins. When inducible chromosomal AmpC is derepressed, the only beta-lactams preserving phenotypic (and clinical) activity are cefepime, carbapenems, and most newer BL-BLICs [3,12,31,32].

(2) Non-inducible chromosomal ampC genes are present in the genomes of some very frequently encountered Enterobacterales, particularly *E. coli* but also *Shigella* spp. They determine the production of a very small quantity of AmpC enzymes, which do not produce a phenotypical or clinically relevant effect on beta-lactam activity [3,17,33,34]. However, some mutations in the regulatory system of these genes (promotor and/or attenuator mutations) can determine their stable overexpression/derepression, resulting in AmpC non-inducible and non-reversible hyperproduction that is independent of beta-lactam exposure and confers resistance to third-generation cephalosporins. This mechanism remains rare [3,35].

(3) Plasmid-encoded ampC genes are derived from inducible chromosomal genes, which have been mobilized on plasmids, with subsequent spreading among various organisms. Plasmid-encoded AmpCs are non-inducible and non-reversible. They have been found in *Escherichia coli*, *Klebsiella pneumoniae*, *Klebsiella oxytoca*, *Proteus mirabilis*, *Salmonella enterica*, and *Shigella* spp. [2,16,17,36,37].

Non-inducible but overexpressed/derepressed chromosomal AmpCs (group 2) and plasmid-encoded AmpCs (group 3) confer phenotypical resistance to third-generation cephalosporins, such as ceftriaxone, cefotaxime, and ceftazidime [3,12]. These resistance mechanisms can have a relevant impact on the appropriateness of empirical treatments from a stewardship standpoint, but they do not create special challenges in interpreting phenotypic antibiotic susceptibility testing (AST) once the isolate is available, since AST is reliable in guiding antibiotic choice.

Inducible chromosomal AmpCs (group 1) are a more relevant and challenging phenomenon. When this resistance mechanism is present, but in the absence of derepression, third-generation cephalosporins display phenotypical susceptibility. During antibiotic treatment, derepression of the ampC gene can result in an increased production of AmpCs, with the emergence of clinically relevant antibiotic resistance to third-generation cephalosporins (Table 1). As a result, AST can be misleading, as it may initially show susceptibility to ceftriaxone, cefotaxime, and ceftazidime, but resistance to these agents can emerge during antibiotic treatment, leading to clinical failure [3,22,23,24,25,26,27,28,32,38].

Because of these considerations, the next paragraphs will focus particularly on inducible chromosomal AmpC-producing Enterobacterales and their treatment.

### 2.3. Which Enterobacterales Produce Inducible Chromosomal AmpCs?

Inducible chromosomal AmpCs can be produced by a variety of Enterobacterales. However, the probability of derepression during antibiotic treatment and the degree of derepression (i.e., the amount of enzyme produced) can differ significantly from one species to another.

*Enterobacter cloacae complex, Klebsiella aerogenes* (formerly known as *Enterobacter aerogenes*), and *Citrobacter freundii* are the Enterobacterales at higher risk for derepression [3,32]. However, the exact risk of derepression during treatment with third-generation cephalosporins is unknown. In some observational studies, it has been estimated at an average of 20% (8% to 40%), but these data require further validation [27,28,29,30,39,40].

Other Enterobacterales that can harbor inducible chromosomal AmpCs are *Serratia marcescens, Providencia spp., Morganella morganii, Hafnia alvei*, and *Yersinia enterocolitica* [3,32]. The exact rate of derepression is unknown for these pathogens as well, but it is estimated to be <5% [3,27,38,39,41]. Furthermore, the derepression of the ampC gene determines a production of AmpC that is up to 10 times less abundant than for *Enterobacter cloacae complex, Klebsiella aerogenes*, and *Citrobacter freundii* [24,39,42].

### 2.4. Which Antibiotics Can Induce AmpC Derepression?

When an inducible chromosomal AmpC is present, some antibiotics can induce AmpC derepression more efficiently than others. On the other hand, some antibiotics are more susceptible to AmpC-mediated hydrolysis than others. According to these characteristics, beta-lactams can be classified into four groups (Table 2) [3,12,22,27,28,29,30,43,44,45,46]:Potent inducers, susceptible to hydrolysis: aminopenicillins (ampicillin, amoxicillin), amoxicillin/clavulanic acid, first-generation cephalosporins, and cephamycins (cefoxitin) are potent inducers of AmpC and are rapidly hydrolyzed by AmpC, even when inducible chromosomal AmpC is not derepressed.Weak inducers, susceptible to hydrolysis: third-generation cephalosporins, piperacillin/tazobactam, and aztreonam are weak inducers but are substrates of AmpC. Increased minimal inhibitory concentrations (MICs) are observed in the absence of AmpC derepression, and clinically relevant resistance is present in cases of derepression.Potent inducers, resistant to hydrolysis: carbapenems are potent inducers, but like cefepime, they remain resistant to hydrolysis through the formation of a stable acyl/enzyme complex.Weak inducers, resistant to hydrolysis: cefepime is a weak AmpC inducer and has the advantage of withstanding AmpC hydrolysis through the formation of a stable acyl/enzyme complex that determines low enzyme affinity and low hydrolysis rate.

## 3. Treatment

The treatment of AmpC-producing Enterobacterales is not well established. Only one small, pilot randomized clinical trial (RCT), showing conflicting results, is available to date [47]. The remaining evidence to guide treatment comes from extrapolation from in vitro data, or observational studies, which suffer from several limitations, including small sample size (limiting the ability to show significant differences between groups), heterogeneous populations, heterogeneous treatment strategies, heterogeneous outcome measures, and high risk of bias due to the inherent impossibility to control for confounders [32,48,49,50,51,52,53,54,55,56,57,58,59,60,61,62,63,64,65,66].

Despite these limitations, the Infectious Diseases Society of America (IDSA) released in 2023 the updated version of the Guidance on the treatment of antimicrobial-resistant Gram-negative infections, including a chapter on the treatment of AmpC-producing Enterobacterales [32]. In contrast, the European Society of Clinical Microbiology and Infectious Diseases (ESCMID) provided only some general indications about the use of cefepime and carbapenems in these conditions in the Guidelines for the treatment of infections caused by MDR Gram-negative bacilli, without issuing any specific recommendations [67].

In order to summarize the best available evidence on antibiotic treatment of AmpC-producing Enterobacterales, it is worthwhile to distinguish between species at moderate-to-high risk of derepression (*Enterobacter cloacae complex, Citrobacter freundii*, and *Klebsiella aerogenes*) and species at low risk of derepression (*Serratia marcescens*, *Providencia* spp., *Morganella morganii*, *Hafnia alvei*, and *Yersinia enterocolitica*).

### 3.1. How to Treat Species at Moderate-to-High Risk of AmpC Derepression?

Carbapenems are considered the reference treatment for Enterobacterales at moderate-to-high risk of AmpC derepression (*Enterobacter cloacae complex*, *Citrobacter freundii*, and *Klebsiella aerogenes*) (Table 3). They are not susceptible to AmpC-mediated hydrolysis, remaining active even in cases of AmpC derepression. Carbapenems preserve their activity in cases of high-inoculum infections or if other additional resistance mechanisms are present, such as ESBLs [68]. Meropenem and imipenem/cilastatin are the reference carbapenems [32,48,49,50,51,52,53,54,55,56,57,58,59,60,62,63,65,69,70], but ertapenem has also been investigated and may be a valid option [64,71]. However, susceptibility to ertapenem needs to be confirmed and cannot be directly inferred from that of meropenem, as ertapenem resistance due to the combination of AmpC hyperproduction and porin loss in ESBLs has been documented [72,73].

Despite the reliable antibacterial activity of carbapenems, carbapenem-sparing regimens are desirable from an antibiotic stewardship perspective. Thus, other narrower-spectrum antibiotics should be considered when AST shows phenotypical susceptibility.

#### 3.1.1. Third-Generation Cephalosporins

The use of third-generation cephalosporins (3GCs: ceftriaxone, cefotaxime, ceftazidime) for the treatment of *Enterobacter cloacae complex*, *Klebsiella aerogenes*, and *Citrobacter freundii* infections is generally discouraged in the case of invasive infections since it is associated with a clinically relevant risk (5–40%) of AmpC derepression and subsequent treatment failure (Table 3) [32,65,74]. In a recent retrospective observational study using a propensity score-based analysis, Maillard et al. included 575 patients with bloodstream infection (BSI) or pneumonia due to AmpC-producing Enterobacterales. They compared patients treated with ceftriaxone or piperacillin/tazobactam (investigational groups) to patients treated with cefepime or meropenem (reference group). They found that the likelihood of clinical failure was higher in the patients treated with ceftriaxone or piperacillin/tazobactam, even if the 30-day mortality was not significantly different [65]. Clinical failure associated with ceftriaxone treatment was also found in another observational study on pulmonary infections [75]. Third-generation cephalosporins retain a role, together with non-beta-lactam options, for the treatment of non-invasive infections, such as uncomplicated urinary tract infections (UTIs) (Table 3) [32,41,76].

However, even if this paradigm is widely shared, some areas of uncertainty remain. Evidence from well-designed RCTs is lacking, and data from observational studies are not univocal in showing an increased risk of clinical failure compared with cefepime or carbapenems [77]. Moreover, several studies did not investigate on a molecular basis whether concomitant ESBL production was present when clinical failure with 3GCs was described. Finally, many studies on the use of ceftriaxone were based on the old CLSI breakpoints (pre-2010: ceftriaxone susceptibility for MICs ≤ 8 mg/L), making the translation of evidence challenging when the current EUCAST and CLSI ceftriaxone susceptibility breakpoints are considered (≤1 mg/L) [77,78,79]. Thus, from an antibiotic stewardship perspective, the role of 3GCs deserves further investigation.

#### 3.1.2. Cefepime

Cefepime is a weak inducer of AmpC derepression, and it is stable against AmpC’s hydrolytic activity. Moreover, unlike other cephalosporins, cefepime is a zwitterion, and it can cross the bacterial outer membrane more quickly, thus being less exposed to the action of β-lactamases [12,80,81].

Because of these biochemical and pharmacological properties, and according to clinical data [32,48,49,50,52,53,54,55,56,57,64,66], cefepime is considered a reliable option for the treatment of invasive infections resulting from AmpC-producing Enterobacterales (Table 3).

However, evidence regarding the clinical use of cefepime when treating cefepime-susceptible AmpC-producing Enterobacterales relies only on observational studies. Overall, when compared to carbapenems, cefepime seems to perform well, without any clear evidence of worse outcomes, but evidence from RCTs is lacking [32,48,49,50,52,53,55,56,57,58,64,66,69,70]. Tamma et al. included patients with BSI, respiratory, and intra-abdominal infections caused by *Enterobacter cloacae complex, Citrobacter freundii*, and *Serratia marcescens*. They showed similar 30-day mortality (31% for cefepime and 34% for carbapenems) in a propensity score-matched cohort (32 patients for each group, after balancing with propensity score matching) [69]. In a well-balanced population of 144 patients, Lee et al. showed the non-inferiority of cefepime (72 patients) compared to meropenem (72 patients) when treating *Enterobacter cloacae* BSI. In the small subgroup of 18 patients having an MIC for cefepime >2 mg/L, there was a higher risk of death [50]. Similar findings were reported in other observational studies [32,48,49,52,53,55,56,57,58,64].

A high dose, i.e., 6 g per day (in 3 administrations or continuous infusion), should be preferred for this indication since a twice-daily dose has been associated with an increased risk of clinical failure [50,58,82].

It is worthwhile to highlight that a discrepancy exists between EUCAST and CLSI MIC breakpoints for cefepime in Enterobacterales. The MIC breakpoint for cefepime susceptibility is ≤1 mg/L for EUCAST and ≤2 mg/L for CLSI; the breakpoint for resistance is >4 mg/L for EUCAST and ≥16 mg/L for CLSI, with an MIC of 4–8 considered susceptible dose-dependent (SDD) [78,79]. The epidemiological cut-off (ECOFF) is 0.125 mg/L according to EUCAST [83]. The more conservative breakpoints proposed by EUCAST appear to be more cautious, particularly in cases of severe infection. In fact, an MIC ≤ 1 mg/L is generally considered a reliable phenotypic proxy to rule out the co-presence of other resistance mechanisms, such as ESBLs or porin mutations, which may affect cefepime efficacy [32,50,84,85,86]. When the MIC is in the range of 4 to 8 mg/l, other mechanisms of resistance cannot be ruled out; there is a higher risk of clinical failure with cefepime, and carbapems become a safer treatment option [3,32,50,53,87]. Real AmpC-mediated resistance (i.e., without the co-participation of other resistance mechanisms) has been reported because of structurally modified AmpC resulting from ampC gene mutations; however, this phenomenon has been rarely described [88,89,90].

The use of high cefepime doses has to be weighed against the risk of adverse reactions. Cefepime-related neurotoxicity is a topic of growing interest, and it is usually associated with renal dysfunction and high trough concentrations [91,92]. Cefepime therapeutic drug monitoring has been suggested to improve cefepime’s safety profile in patients at risk of toxicity [93].

#### 3.1.3. Piperacillin/Tazobactam

Tazobactam is vulnerable to AmpC-mediated hydrolysis, but piperacillin/tazobactam is a weak inducer of AmpC derepression, making the probability of clinical failure when treating a piperacillin/tazobactam-susceptible strain theoretically low [12]. Consequently, piperacillin/tazobactam is considered a possible option for the treatment of wild-type *Enterobacter cloacae complex*, *Klebsiella aerogenes*, and *Citrobacter freundii* (Table 3).

A small, pilot RCT (MERINO 2) enrolling 72 patients compared piperacillin/tazobactam versus meropenem for definitive treatment of bloodstream infections caused by *Enterobacter* spp., *Citrobacter freundii*, *Serratia marcescens*, *Morganella morganii*, and *Providencia* spp. The study considered a composite outcome, including death, clinical failure, microbiological failure, and microbiological relapse at 30 days [47]. The study did not show a statistically significant difference in the piperacillin/tazobactam arm versus the meropenem arm, with 28% and 21% of patients meeting the composite outcome, respectively (risk difference: 8%; 95% confidence interval: −12 to 28%). However, the study included only 38 and 34 patients for each arm, and no sample size calculation was performed; thus, it was very probably underpowered to detect meaningful differences between the two groups. Moreover, when the authors considered the subcomponents of the primary outcome, microbiological failure was significantly more probable in the piperacillin/tazobactam group. In conclusion, the results of the trial have to be considered as purely exploratory, and further studies are needed to confirm their validity [47].

Data from observational studies are conflicting. Some studies showed that piperacillin/tazobactam was non-inferior to carbapenems, but several limitations apply, including small sample size and imbalanced populations [51,53,54,66,70,94]. Cheng et al. performed a propensity score-matched analysis, including 41 patients in each group (piperacillin/tazobactam versus cefepime or meropenem), showing no statistically significant differences in 30-day mortality. However, there was a trend toward increased mortality in the piperacillin/tazobactam group (15% versus 7%), and the small sample size raises concerns about a possible type II error. Chaubey et al. came to some different conclusions, showing that piperacillin/tazobactam was inferior to cefepime and to meropenem in terms of in-hospital mortality; however, the study was limited by a small sample size as well. In the aforementioned larger observational study, Maillard et al. found that ceftriaxone or piperacillin/tazobactam treatment had a higher risk of clinical failure (but similar 30-day mortality) compared to cefepime or meropenem as a control [65].

In terms of ecological impact (selective pressure) and risk for MDR selection, piperacillin/tazobactam and cefepime are probably comparable [95]. Piperacillin/tazobactam has been associated with a higher risk of nephrotoxicity, particularly when associated with other nephrotoxic drugs, such as vancomycin [96,97,98,99], but this difference was not confirmed in a recent RCT [92]. Cefepime has been associated with a higher rate of drug-induced neurotoxicity [92].

IDSA guidelines suggest a preference for cefepime or carbapenems over piperacillin/tazobactam for severe infections caused by AmpC-producing Enterobacterales. Piperacillin/tazobactam remains a viable option for non-severe infections, such as UTIs; its role in cases of severe infection deserves further investigation [32,76].

#### 3.1.4. Ceftolozane/Tazobactam, New Beta-Lactam/Beta-Lactamase Inhibitor Combinations, and Cefiderocol

Ceftolozane/tazobactam has shown good activity against AmpC-producing *Pseudomonas aeruginosa*, unless an AmpC-Ω-loop mutation is present. This is due to the intrinsic resistance of ceftolozane to pseudomonal AmpC-mediated hydrolysis [5]. In contrast, data about the activity of ceftolozane/tazobactam against AmpC-producing Enterobacterales are scarce and less encouraging, making this drug a non-preferred choice for this indication (Table 3) [32,73,100].

New commercially available beta-lactam/beta-lactamase inhibitor combinations (BL-BLICs, i.e., ceftazidime/avibactam, imipenem/cilastatin/relebactam, meropenem/vaborbactam) and cefiderocol have shown good activity in vitro against AmpC-producing Enterobacterales [101,102,103]. However, the available clinical data are very limited and focused mainly on ceftazidime/avibactam [104,105]. Moreover, the emergence of resistance in AmpC-producing Enterobacterales has been described for ceftazidime/avibactam and cefiderocol, even if it probably remains very rare [89,106]. Because of these elements, and from an antibiotic stewardship perspective, it is advisable to reserve the use of new commercially available BL-BLICs and cefiderocol for the treatment of carbapenem-resistant strains (Table 3) [32,101].

#### 3.1.5. Non-Beta-Lactam Antibiotics

Non-beta-lactam antibiotics do not induce AmpC derepression and are not susceptible to AmpC-mediated hydrolysis. They can be considered for two main indications: as first-line agents or as oral step-down therapy [32,70]. Fluoroquinolones and trimethoprim/sulfamethoxazole are the most relevant non-beta-lactam antibiotics in this context, particularly because of their clinical efficacy and high oral bioavailability [32,107].

Fluoroquinolones and trimethoprim/sulfamethoxazole can be considered for uncomplicated UTIs [3,32,107,108,109]. Moreover, due to their good intra-prostatic penetration, they are considered drugs of choice for both acute and chronic prostatitis [109,110]. Nitrofurantoin can also play a role in uncomplicated female cystitis [109,111]. In contrast, the use of fosfomycin for uncomplicated female cystitis should be mainly reserved for infections caused by *Escherichia coli* [112].

In recent years, there has been growing interest in early oral step-down treatment for several serious infections, such as uncomplicated BSI [113,114,115,116,117,118], infective endocarditis [119], and bone and joint infections (BJIs) [120]. Evidence showed that oral step-down can be as effective as intravenous treatment when patients are appropriately selected, with a decrease in the duration of hospital stay [113,114,115,116,117,118,119]. Fluoroquinolones and trimethoprim/sulfamethoxazole play a major role in this context because of their excellent bioavailability, particularly when treating Gram-negative uncomplicated BSI with a urinary source [113,114,118,121]. Major studies included mostly infections caused by *Escherichia coli* and *Klebsiella pneumoniae*, with relatively few AmpC-producing Enterobacterales [114,115]. A partial exception was the retrospective multicenter study by Tamma et al., which included a propensity score-matched cohort of 4967 unique patients hospitalized with monomicrobial BSI caused by Enterobacterales. In the study, *Enterobacter* spp. and *Citrobacter* spp. accounted for around 15% of included patients in total [118].

In summary, data about the efficacy and safety of partial oral treatment of Gram-negative infections, particularly uncomplicated BSI caused by Enterobacterales (including AmpC-producing strains), are solid and encouraging, supporting oral step-down as a reliable strategy [32,113]. The best evidence is available for fluoroquinolones and, to a lesser extent, for trimethoprim/sulfamethoxazole [114,115,116,117,118].

### 3.2. What Is the Treatment for Species at Low Risk of AmpC Derepression?

Some species considered at low risk of derepression are *Serratia marcescens, Providencia* spp., *Morganella morganii, Hafnia alvei*, and *Yersinia enterocolitica*.

As already underlined in Section 2.3, the real risk of derepression is unknown, but it is estimated to be <5% [3,27,38,39,41]. Furthermore, AmpC gene derepression led to a much less relevant production of AmpC enzyme (up to 10 times less) than in *Enterobacter cloacae complex, Klebsiella aerogenes*, and *Citrobacter freundii* [24,39,42].

Once confirmed that the strain does not carry any ESBL enzyme and the MIC for 3GCs falls into the wild-type range, it seems reasonable to propose ceftriaxone or cefotaxime as first-line therapy, as proposed by the IDSA guidelines [32,122]. Some concerns may arise in cases of high-inoculum infection, particularly when source control is unfeasible or in cases of very severe infections. In these carefully selected cases, some may consider cefepime or carbapenems as a more cautious choice. However, the risk/benefit ratio of such a choice is unclear, particularly for infections needing prolonged treatment (i.e., osteoarticular infections or complex, undrainable abscesses), where a prolonged exposure to antibiotics is unavoidable [41].

Concerning difficult-to-treat infections, it is worthwhile to remember that *Serratia marcescens* has been categorized as a typical pathogen for prosthetic valve endocarditis, according to the 2023 Duke-International Society for Cardiovascular Infectious Diseases Criteria for Infective Endocarditis [123]. The 2023 European Society of Cardiology Guidelines for the Management of Endocarditis suggested a combination treatment for endocarditis caused by non-HACEK Gram-negative pathogens, including *Serratia marcescens* (beta-lactam plus aminoglycosides or fluoroquinolones), but this approach is not well supported by clinical evidence [124,125,126].

## 4. Future Perspectives

The treatment of AmpC-producing Enterobacterales is a challenging and evolving field. Many aspects deserve further investigation, including the following:(1)The exact risk of derepression in different species and related to different antibiotics (including newer molecules, such as ceftolozane/tazobactam, new BL-BLICs, and cefiderocol);(2)The clinical efficacy of 3GCs and piperacillin/tazobactam, depending on the clinical context and causative pathogen (low or moderate-to-high risk of derepression). This would ideally require data from well-conducted RCTs;(3)The role of cefepime and ertapenem, compared to meropenem or imipenem/cilastatin, in difficult-to-treat infections (e.g., BJIs), high-inoculum infections (e.g., undrainable abscesses), and very severe infections (e.g., septic shock);(4)The role of ceftolozane/tazobactam;(5)The role of non-beta-lactam options beyond UTIs and oral step-down;(6)The eventual need for escalation when the clinical isolate becomes available but the patient has already achieved clinical stability and has been empirically treated with second-line agents, such as 3GC;(7)The role of de-escalation to 3GC (particularly for prolonged treatment courses) when the patient has already reached clinical stability and is not eligible for oral step-down.

## Figures and Tables

**Figure 1 pharmacy-12-00142-f001:**
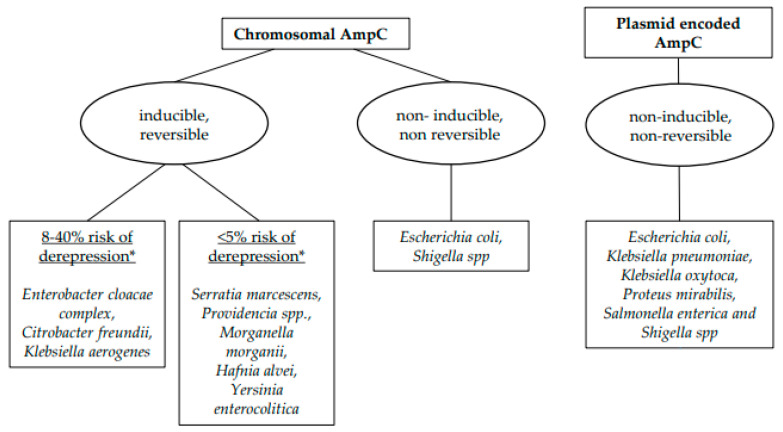
Classification of AmpC enzymes produced by Enterobacterales. * During antibiotic treatment with beta-lactams (each beta-lactam is associated with a different risk of derepression).

**Table 1 pharmacy-12-00142-t001:** AmpC-producing Enterobacterales: expected phenotypes for wild-type strains and in cases of AmpC derepression and ESBL production.

Antibiotic	Wild-Type Expected Phenotype	Expected Phenotype in Cases of AmpC Derepression	Expected Phenotype in Cases of ESBLs
aminopenicillins (ampicillin, amoxicillin)	R	R	R
Amoxicillin/clavulanic acid	R	R	S/R ^3^
ticarcillin	S	R	R
piperacillin	S	R	R
piperacillin/tazobactam	S	R	S/R ^3^
first-generation cephalosporins	R	R	R
cephamycins (2ndG cef)	R	R	S
third-generation cephalosporins	S	R	R
fourth-generation cephalosporins (cefepime)	S	S ^1^	S/R ^3^
ceftaroline and ceftobiprole	S	R	R
ceftolozane/tazobactam	S	S/R	S/R ^4^
aztreonam	S	R	S/R ^3^
carbapenems	S	S ^2^	S ^2^
ceftazidime/avibactam, imipenem/cilastatin/relebactam, meropenem/vaborbactam, cefiderocol	S	S	S

ESBLs: extended-spectrum beta-lactamases. ^1^: Expected MIC: ≤1 mg/L in absence of ESBL co-production; >2 in presence of ESBL co-production. ^2^: Susceptibility to ertapenem has to be tested and cannot be inferred from susceptibility to meropenem or imipenem/cilastatin. ^3^: In case of susceptibility, MICs will not be in wild-type range. ^4^: Resistance to ceftolozane/tazobactam due to ESBL production is infrequent.

**Table 2 pharmacy-12-00142-t002:** Ability to induce ampC gene derepression and subsequent AmpC production and susceptibility to AmpC-mediated hydrolysis of different beta-lactams.

		Ability to Induce AmpC Gene Derepression and Subsequent AmpC Production	
		Low	High
Susceptibility to AmpC-mediated hydrolysis	low	cefepime	carbapenems
	high	third-generation cephalosporins, piperacillin/tazobactam, aztreonam	aminopenicillins (ampicillin, amoxicillin), amoxicillin/clavulanic acid, first-generation cephalosporins, cephamycins (cefoxitin)

**Table 3 pharmacy-12-00142-t003:** Microbiological and clinical properties of the most relevant beta-lactams used for treating infections caused by AmpC-producing Enterobacterales with moderate-to-high risk of AmpC derepression (*Enterobacter cloacae, Klebsiella aerogenes*, and *Citrobacter freundii*).

Antibiotic	Substrate for AmpC-Mediated Hydrolysis	Induction of AmpC Derepression	Clinical Data	Clinical Use	Antibiotic Stewardship Additional Considerations
Third-generation cephalosporins	Yes	Strong inducers	Observational studies	Discouraged, except for low-risk UTIs	Good choice in cases of AmpC-producing Enterobacterales with low risk of derepression ^1^, except for very severe infections
Cefepime	No ^2^	Weak inducer	Observational studies	Drug of choice. Non-inferiority to carbapenems has not been demonstrated in RCTs	Preferable to carbapenems from a stewardship perspective
Piperacilin/tazobactam	Yes	Weak inducer	One pilot RCT + observational studies	More commonly considered as a second choice compared to cefepime and carbapenems. Probably avoided for more severe and high-inoculum infections	Probably comparable to cefepime in terms of selective pressure (ecological impact)
Ceftolozane/tazobactam	Insufficient evidence	Insufficient evidence	Insufficient evidence (mainly in vitro data)	Not indicated except when treating polymicrobial infections, including *P. aeruginosa*	The exact extent of its ecological impact is unknown but should theoretically be less relevant than for carbapenems. Use can be limited also by high cost
Ertapenem	No ^3^	Strong inducer (? few data)	Few observational studies	Reasonable alternative to meropenem and imipenem/cilastatin	Cefepime is preferable from a stewardship perspective
Imipenem/cilastatin and meropenem	No	Strong inducer	Observational studies, one pilot RCT. Usually used as comparators for other investigational drugs	Drug of choice	
New BL-BLICs and cefiderocol	No ^4^	Insufficient evidence	Few observational studies	Drugs of choice for carbapenem-resistant strains	To be reserved for carbapenem-resistant strains

^1^: *Serratia marcescens, Providencia* spp., *Morganella morganii*, *Hafnia alvei*, *Yersinia enterocolitica*. ^2^: Stable against AmpC-mediated hydrolysis. True AmpC-mediated resistance has been described because of mutated AmpC. Otherwise (and more commonly), cefepime resistance is determined by coexistence of AmpCs and ESBLs or porinmutations. ^3^: Resistance has been described when AmpC derepression and porin mutations coexist. ^4^: Resistance to ceftazidime/avibactam and cefiderocol had been associated with mutated AmpC. BL-BLICs: beta-lactam/beta-lactamase inhibitor combinations; RCT: randomized controlled trial; UTI: urinary tract infection.

## Data Availability

Not applicable.

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
