# Peer review of "Antibiotic Treatment of Infections Caused by AmpC-Producing Enterobacterales"

_pharmacy, 2024, doi:10.3390/pharmacy12050142_

Round 1
Reviewer 1 Report
Comments and Suggestions for Authors
This comprehensive review covers the microbiological properties on AmpC- producing Enterobacterales, and the capacity of several therapeutic agents targeting these bacteria.
The manuscript is well-written, and it provides an overall picture about the most relevant information in this area of knowledge. For these reasons, we consider that it deserves to be published in Pharmacy with minor revisions:
Some figures could be added to improve the overall presentation. Also, the Conclusions section could be better written to improve the overall soundness of the work.
Reviewer 2 Report
Comments and Suggestions for Authors
In their ms, the authors have reviewed relevant literature on antimicrobials for combatting AmpC-producing enterobacteria, and this topic falls to scopes of Pharmacy journal. However, the manuscript has roughly been written and needs serious editing and improvement in view of numerous problems with writing style and terminological problems, especially in microbiology, and spelling errors. English and typing should substantially be improved.
Major Comments
· “Antibiotic treatment of “Enterobacteriales” as in title and of any bacteria or species as elsewhere in the text must be avoided and replaced. One can treat diseases caused by bacterial infections, one can treat bacterial suspensions with antimicrobials, one can try to kill bacterial by using antibiotics. Such jargons, like antibiotic treatment of bacteria, are not appropriate in the scientific literature.
· It will be helpful to summarize briefly the available reviews on this topic and to highlight the rationale for writing a new review.
· Subsection 2.2: As for the paragraph (1), I recommend elaborating information on the AmpC induction mechanism: it is complex and should be described clearly.
· Table 1 is difficult for understanding; it has been named as “ AmpC-producing Enterobacterales” Where are enterobacteria?
· Table 3. Synthesis of microbiological and clinical properties … This is not synthesis.
· I recommend avoiding a connotation of synthesis even in its philosophic meaning
· Enterobacterales, not Enterobacteriales, is the approved name of the order. Such misspelling is inadmissible
· “Gram-negative bacilli” should be avoided; a term “bacilli” is also used for the representatives of Bacillus genus and they are Gram-negative.
· A title “Which is the treatment of species at low risk of AmpC derepression?” must be revised.
· Concluding remarks have been written in too technically and uniformly. Thus, “a better understanding of the clinical efficacy/risk” appears several times.
· A final conclusions that “ further studies are needed to improve clinical management of AmpC-producing Enterobacterales” is banal and brings nothing .
· Misspelling errors must be checked: where(were), synthetize, and others
· Which should be replaced in titles with What or Which of.
· Numerous cumbersome sentences should be simplified.
Comments on the Quality of English LanguageEnglisn needs extensive editing and improments in readability. Some my comments are given in Comments to Authors.
Reviewer 3 Report
Comments and Suggestions for Authors
The manuscript of Tebano et al. review the most relevant information about microbiological properties of ampC-producing Enterobacterales and their possible therapeutic approaches . The manuscript describe the challenging clinical practice for the treatment of AmpC-producing Enterobacterales and highlights the need of further studies to improve the clinical management of these infections.
Minor comments:
Table 1A and B: Transform the 2 tables in 1, adding a column for the wild-type phenotype and another for the AmpC derepression phenotype. In my opinion is easier to see the resistance profile difference.
Section 2.2: I miss some image illustrative representations in the manuscript. I suggest an image to illustrate the 3 AmpC groups.
Section 3.1.1: Please add the meaning of abbreviation BSI.
Round 2
Reviewer 2 Report
Comments and Suggestions for Authors
No further comments